# An adjunctive therapy administered with an antibiotic prevents enrichment of antibiotic-resistant clones of a colonizing opportunistic pathogen

Valerie J Morley[1]*, Clare L Kinnear[2], Derek G Sim[1], Samantha N Olson[1], Lindsey M Jackson[1], Elsa Hansen[1], Grace A Usher[3], Scott A Showalter[3,4], Manjunath P Pai[5], Robert J Woods[2], Andrew F Read[1,6,7]

[1]Center for Infectious Disease Dynamics, Department of Biology, The Pennsylvania State University, University Park, United States; [2]Division of Infectious Diseases, Department of Internal Medicine, University of Michigan, Ann Arbor, United States; [3]Department of Biochemistry and Molecular Biology, The Pennsylvania State University, University Park, United States; [4]Department of Chemistry, The Pennsylvania State University, University Park, United States; [5]Department of Clinical Pharmacy, College of Pharmacy, University of Michigan, Ann Arbor, United States; [6]Huck Institutes for the Life Sciences, The Pennsylvania State University, University Park, United States; [7]Department of Entomology, The Pennsylvania State University, University Park, United States

*For correspondence:
vum84@psu.edu

**Competing interests:** The authors declare that no competing interests exist.

**Abstract** A key challenge in antibiotic stewardship is figuring out how to use antibiotics therapeutically without promoting the evolution of antibiotic resistance. Here, we demonstrate proof of concept for an adjunctive therapy that allows intravenous antibiotic treatment without driving the evolution and onward transmission of resistance. We repurposed the FDA-approved bile acid sequestrant cholestyramine, which we show binds the antibiotic daptomycin, as an 'anti-antibiotic' to disable systemically-administered daptomycin reaching the gut. We hypothesized that adjunctive cholestyramine could enable therapeutic daptomycin treatment in the bloodstream, while preventing transmissible resistance emergence in opportunistic pathogens colonizing the gastrointestinal tract. We tested this idea in a mouse model of *Enterococcus faecium* gastrointestinal tract colonization. In mice treated with daptomycin, adjunctive cholestyramine therapy reduced the fecal shedding of daptomycin-resistant *E. faecium* by up to 80-fold. These results provide proof of concept for an approach that could reduce the spread of antibiotic resistance for important hospital pathogens.

## Introduction

Vancomycin-resistant *Enterococcus faecium* (VR *E. faecium*) is an important cause of antibiotic-resistant infections in healthcare settings (*Arias and Murray, 2012*; *García-Solache and Rice, 2019*; *O'Driscoll and Crank, 2015*). The antibiotic daptomycin is one of the few remaining first-line therapies for VRE infection (*O'Driscoll and Crank, 2015*), but daptomycin-resistance is spreading in VRE populations (*Judge et al., 2012*; *Kamboj et al., 2011*; *Kinnear et al., 2019*; *Woods et al., 2018*). Therapeutic daptomycin use is thought to be a key driver of resistance (*Kinnear et al., 2020*; *Woods et al., 2018*). Managing the evolution of daptomycin-resistance in healthcare settings is crucial to future control of VRE infections.

**eLife digest** Antibiotics are essential for treating infections. But their use can inadvertently lead to the emergence of antibiotic-resistant bacteria that do not respond to antibiotic drugs, making infections with these bacteria difficult or impossible to treat. Finding ways to prevent antibiotic resistance is critical to preserving the effectiveness of antibiotics.

Many bacteria that cause infections in hospitals live in the intestines, where they are harmless. But these bacteria can cause life-threatening infections when they get into the bloodstream. When patients with bloodstream infections receive antibiotics, the bacteria in their intestines are also exposed to the drugs. This can kill off all antibiotic-susceptible bacteria, leaving behind only bacteria that have mutations that allow them to survive the drugs. These drug-resistant bacteria can then spread to other patients causing hard-to-treat infections.

To stop this cycle of antibiotic treatment and antibiotic resistance, Morley et al. tested whether giving a drug called cholestyramine with intravenous antibiotics could protect the gut bacteria. In the experiments, mice were treated systemically with an antibiotic called daptomycin, which caused the growth of daptomycin-resistant strains of bacteria in the mice's intestines. In the laboratory, Morley et al. discovered that cholestyramine can inactivate daptomycin.

Giving the mice cholestyramine and daptomycin together prevented the growth of antibiotic-resistant bacteria in the mice's intestines. Moreover, cholestyramine is taken orally and is not absorbed into the blood. It therefore only inactivates the antibiotic in the gut, but not in the blood.

The experiments provide preliminary evidence that giving cholestyramine with antibiotics might help prevent the spread of drug resistance. Cholestyramine is already used to lower cholesterol levels in people. More studies are needed to determine if cholestyramine can protect gut bacteria and prevent antibiotic resistance in people.

*E. faecium* is an opportunistic pathogen that colonizes the human GI tract asymptomatically, spreads via fecal-oral transmission, and causes symptomatic infections when introduced to sites like the bloodstream or the urinary tract (*Arias and Murray, 2012*). *E. faecium* colonizing the gut may be exposed to daptomycin during therapeutic use, potentially contributing to the transmission of daptomycin-resistant *E. faecium*. Daptomycin is administered intravenously to treat infections caused by pathogens including VRE and *Staphylococcus aureus*. Daptomycin is primarily eliminated by the kidneys, but 5–10% of the dose enters the intestines through biliary excretion (*Woodworth et al., 1992*). We hypothesize that this therapeutically unnecessary intestinal daptomycin exposure could drive resistance evolution in *E. faecium* colonizing the gut. Increased resistance in colonizing populations is important, because gut *E. faecium* populations are sources for nosocomial infections and transmission between patients (*Alevizakos et al., 2017*; *Olivier et al., 2008*).

If unintended intestinal daptomycin exposure drives resistance evolution in *E. faecium*, this offers an opportunity to intervene. The opportunity emerges from a key feature of this system—the bacteria causing infection are physically separated from the population contributing to transmission. If daptomycin could be inactivated in the intestine without altering plasma concentrations, daptomycin could be used to kill bacteria at the target infection site without driving resistance in off-target populations. Preventing resistance evolution in these reservoir populations could protect patients from acquiring resistant infections, and it could limit the shedding of resistant strains and so onward transmission to other patients. We hypothesized that co-administering an oral adjuvant that reduces daptomycin activity would prevent selection for daptomycin-resistance in the gut during systemic daptomycin treatment. We tested this strategy using the adjuvant cholestyramine in a mouse VR *E. faecium* gut colonization model.

## Results

### Generation of daptomycin-resistant VR *E. faecium* in the mouse GI tract

To directly test the proposition that systemic daptomycin treatment could select for resistance in the GI tract, and to generate daptomycin-resistant VR *E. faecium* mutants for subsequent experiments, we inoculated mice orally with daptomycin-susceptible VR *E. faecium* strains. Beginning one day

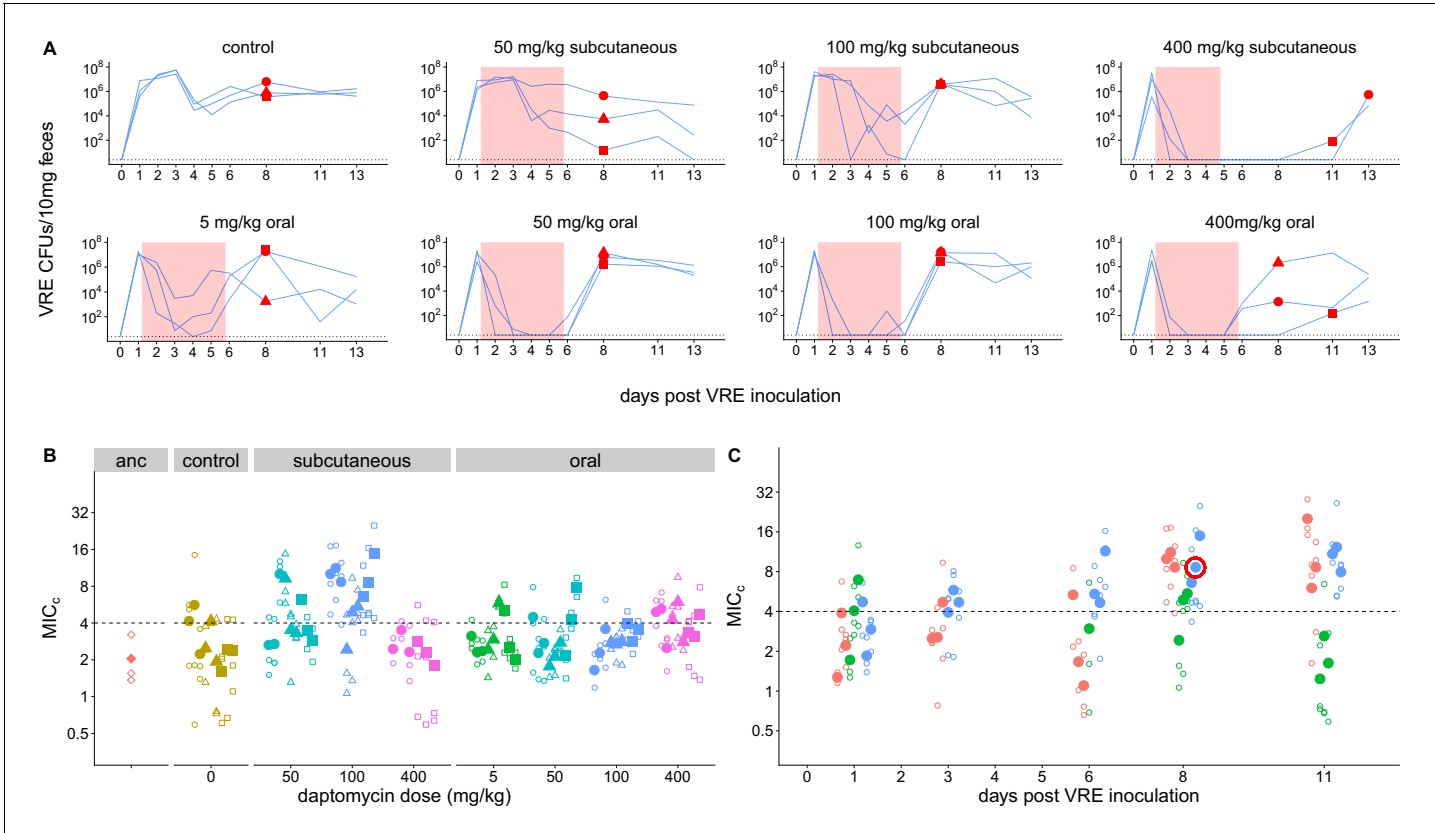

**Figure 1.** Emergence of daptomycin-resistant VR *E. faecium* in mouse GI tracts following subcutaneous daptomycin treatment. (**A**) VR *E. faecium* densities in fecal samples during and after daptomycin treatment (strain BL00239-1). Each line represents VR *E. faecium* densities from an individual mouse (N = 3 per treatment). The pink shaded region indicates days of daptomycin therapy. The dotted line marks the detection limit. Red dots indicate time points where clones were isolated for analysis shown in Panel B. The 400 mg/kg subcutaneous treatment was discontinued after 4 days due to apparent toxicity, and one mouse in this treatment was euthanized at Day 4. (**B**) Following daptomycin treatment, three VR *E. faecium* clones were isolated from the feces of each mouse. Filled points show the mean of triplicate daptomycin susceptibility (MIC$_C$) measurements for each clone, and open points show the individual measurements. Point shape indicates the mouse of origin. The dashed line marks the clinical breakpoint for daptomycin susceptibility. The ancestral clone (BL00239-1) used to inoculate mice is also shown. (**C**) For the 100 mg/kg subcutaneous treatment, VR *E. faecium* clones were isolated from each mouse at multiple time points. Filled points show the mean of triplicate daptomycin susceptibility (MIC$_C$) measurements for each clone, and open points show individual measurements. Color indicates mouse of origin. The dotted line marks the clinical breakpoint for daptomycin susceptibility. The resistant clone used in subsequent experiments (BL00239-1-R) is circled in red.

The online version of this article includes the following source data for figure 1:

**Source data 1.** VR *E. faecium* fecal density data (*Figure 1A*).

**Source data 2.** MIC data (*Figure 1B–C*).

after *E. faecium* inoculation, mice received daily doses of either subcutaneous daptomycin (50, 100, or 400 mg/kg), oral daptomycin (5, 50, 100, or 400 mg/kg), or a control mock injection for 5 days. We used a range of doses and routes of administration to maximize the likelihood of observing resistance emergence in at least one of the mice. The 50 and 100 mg/kg subcutaneous doses were selected to generate pharmacokinetics similar to clinical human doses (*Mortin et al., 2007*; *Samonis et al., 2008*), and the 5 mg/kg oral approximates the 5–10% of a daptomycin dose that is secreted into the intestines during standard intravenous treatment (*Woodworth et al., 1992*). We used two susceptible VR *E. faecium* strains, BL00239-1 (MIC$_c$ = 2.0 (Minimum Inhibitory Concentration computed, see Methods)) and PR00708-14 (MIC$_c$ = 2.7), which were originally isolated at the University of Michigan Hospital from a clinical bloodstream infection and a different patient's clinical perirectal swab, respectively. Mouse fecal samples were collected to quantify VR *E. faecium* shedding and determine daptomycin susceptibility of isolated *E. faecium* clones.

Only very high daptomycin doses (400 mg/kg subcutaneous, ≥50 mg/kg oral) consistently reduced fecal VR *E. faecium* below the level of detection during treatment; with lower doses, VR *E. faecium* shedding was often detectable throughout treatment (*Figure 1A*). For Strain BL00239-1, *E. faecium* clones with increased daptomycin-resistance were isolated from two of three mice following treatment with 100 mg/kg subcutaneous daptomycin (*Figure 1B–C*). We chose one of these daptomycin-resistant clones to use in subsequent experiments (strain BL00239-1-R, $MIC_c$ = 8.6). Sequencing of the core genome showed that the resistant strain acquired a mutation in the major cardiolipin synthase *clsA* gene (R211L, CGA→CTA), which has been previously described in association with daptomycin-resistance (*Adams et al., 2015*), and a transposon insertion into the methionine sulfoxide reductase *msrA* gene (*Zhao et al., 2010*). For the second strain (PR00708-14), mice were treated with subcutaneous daptomycin (50, 100, or 200 mg/kg) or a mock injection. We screened for the emergence of increased resistance by plating mouse fecal suspensions on daptomycin-supplemented agar. Samples from two mice treated with 200 mg/kg daptomycin produced colonies on daptomycin-supplemented plates, and we isolated three clones from each of these samples. These isolated clones had increased daptomycin $MIC_c$ relative to PR00708-14 by broth microdilution ($MIC_c$ = 13.5, 11.5, and 8.0 from one mouse; $MIC_c$ = 12.0, 29.8, and 12.0 from the second mouse). We chose one of these isolated clones for use in subsequent experiments (PR00708-14-R, $MIC_c$ = 12.0). Genome sequencing revealed that PR00708-14 and PR00708-14-R differed by two

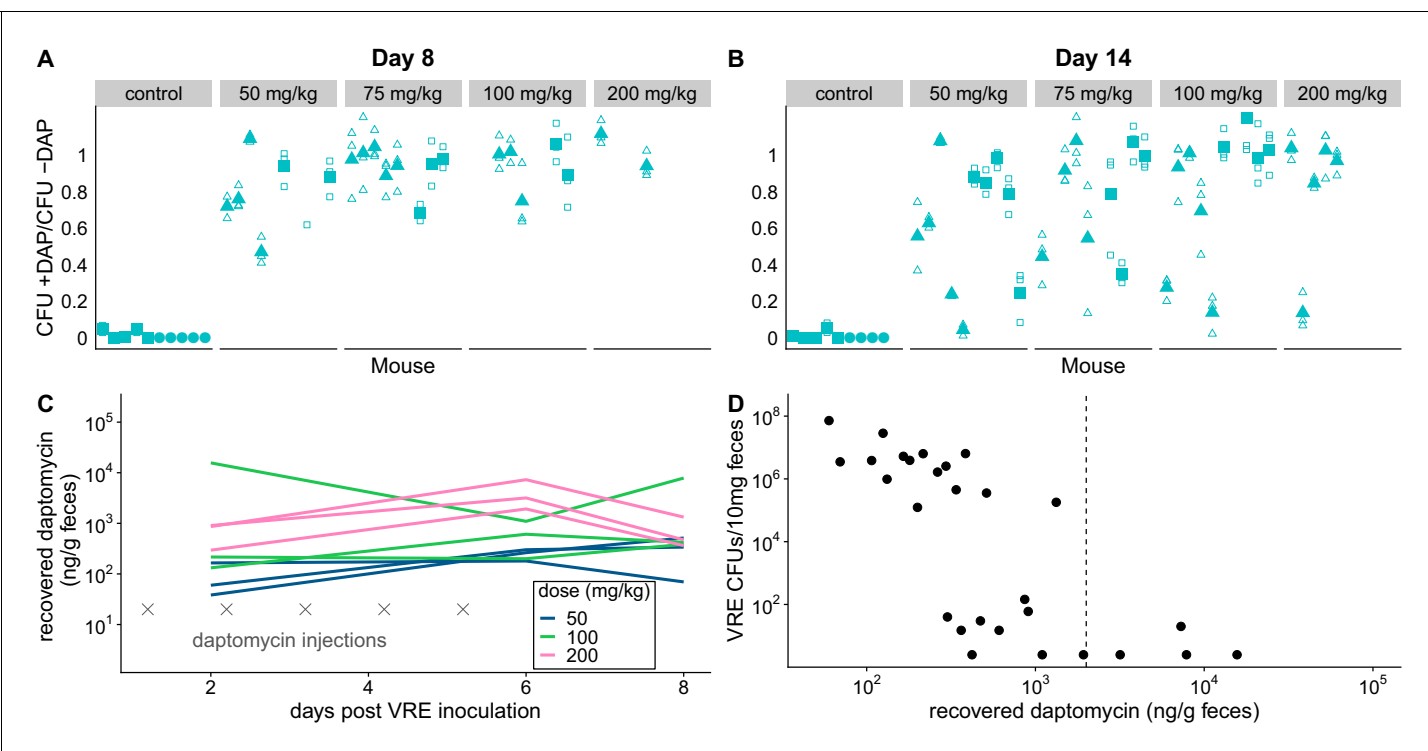

**Figure 2.** Subcutaneous daptomycin treatment selects for resistance in the GI tract. (A-B) Mouse fecal suspensions were plated on *Enterococcus*-selective plates with daptomycin (+ DAP) and without daptomycin (- DAP) at Day 8 (A) and Day 14 (B). Each filled point represents the mean of triplicate measures from a single mouse sample, and open points show individual measurements. Mice were treated with daptomycin for 5 days (triangles) or 10 days (squares) at the doses listed at times denoted (gray crosses in C; N = 5 mice per treatment). Samples with VR *E. faecium* density <3×10³ CFU/10 mg feces were not plated for this assay due to insufficient bacterial density. (C) Recovered fecal daptomycin measured by LC-MS for a subset of mice. Each line tracks daptomycin measurements from a single mouse sampled at Days 2, 6, and 8. (D) For the subset of fecal samples analyzed by LC-MS, fecal daptomycin plotted against fecal VR *E. faecium* densities (samples from all available treatments and time points plotted together). Dotted line indicates the MIC of the susceptible strain BL00239-1 (MIC = 2 μg/mL or 2 μg/g).

The online version of this article includes the following source data and figure supplement(s) for figure 2:

**Source data 1.** VR *E. faecium* daptomycin susceptibility data (*Figure 2A–B*).
**Source data 2.** Fecal daptomycin concentration data (*Figure 2C–D*).
**Figure supplement 1.** Dynamics of VRE shedding for experiment shown in *Figure 2*.

mutations, which to our knowledge have not previously been associated with daptomycin-resistance. We identified mutations in a TerC family integral membrane protein (locus tag HMPREF0351_10759 in D0 *E. faecium* reference genome, I22T, CGA→CTA) and in a hypothetical protein (locus tag HMPREF0351_12146 in D0 *E. faecium* reference genome, frameshift, deleted G at 190nt). These experiments show that daptomycin-resistance can emerge de novo in *E. faecium* colonizing the GI tract following systemic daptomycin treatment.

### Daptomycin treatment enriches for daptomycin-resistant VR E. faecium in the GI tract

We used the de novo resistant mutants isolated above (*Figure 1*) to test whether daptomycin therapy selects for daptomycin-resistance in intestinal VR *E. faecium* populations when a resistant mutant is already present. We orally inoculated mice with a 1:20 mixture of the experimentally generated daptomycin-resistant and susceptible VR *E. faecium* strains (BL00239-1-R and BL00239-1). Note that this approach – seeding inocula with known numbers of resistant bacteria – allows the response to selection to be measured while avoiding the experimental noise introduced by mutation waiting times. Mice were treated with subcutaneous daptomycin (50, 75, 100, or 200 mg/kg) for five or ten days after VRE inoculation. Control mice received either a mock saline injection or no injection. Fecal samples from Days 8 and 14 post-inoculation were plated in triplicate to quantify total VR *E. faecium* density, and samples were also plated on daptomycin-supplemented agar to estimate the proportion of VR *E. faecium* that were daptomycin-resistant. Control populations remained susceptible to

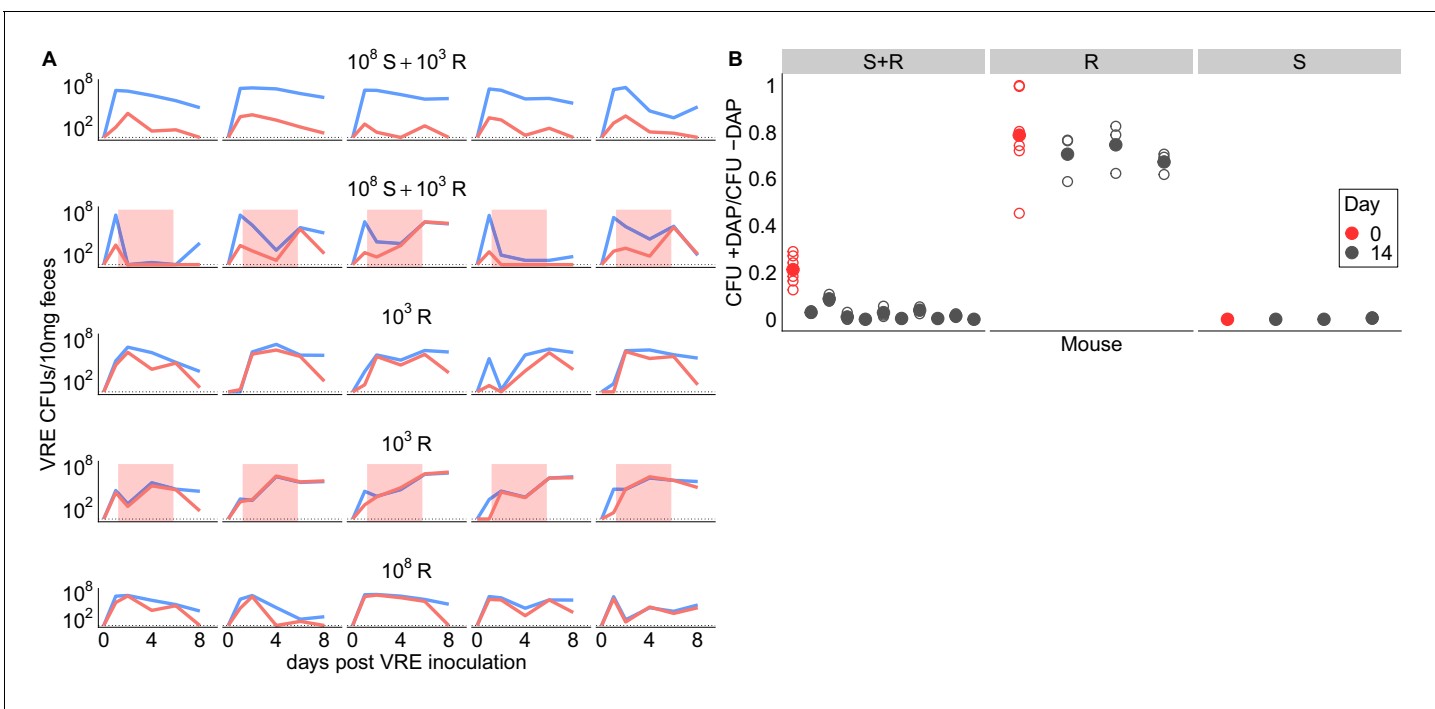

**Figure 3.** Competitive dynamics between daptomycin-resistant and susceptible VRE *faecium*. (**A**) Each panel shows VR *E. faecium* counts on plates with daptomycin (+DAP, red line) and without daptomycin (-DAP, blue line) for a single mouse over time. Labels show initial inocula. Red shading indicates days of daptomycin treatment (100 mg/kg daily subcutaneous injections). (**B**) In a second experiment, mice were inoculated with a mix of susceptible and resistant bacteria (S+R, 20% R), resistant bacteria only (R), or susceptible bacteria only (S). Mice received no drug treatment. At Day 14, fecal suspensions were plated on plates with daptomycin (+DAP) and without daptomycin (-DAP) to determine whether the resistant strain had decreased in frequency. The starting inoculum dose is shown in red, and Day 14 samples from each mouse are shown in gray. Filled points show means of replicate measurements for a sample, and open circles show individual measurements. Note that resistant bacteria do not form colonies at 100% efficiency on +DAP plates.

The online version of this article includes the following source data for figure 3:

**Source data 1.** VR *E. faecium* density data (*Figure 3A*).
**Source data 2.** VR *E. faecium* daptomycin susceptibility data (*Figure 3B*).

daptomycin, but all doses and durations of daptomycin dramatically enriched for resistance in the GI tract (*Figure 2A–B*). At both time points, controls had significantly lower proportions of resistant bacteria than daptomycin-treated mice (*Figure 2A–B*; mixed effects negative binomial regression, p<0.01, see Model one in *Supplementary file 1*). The effect sizes (Cohen's d) at Days 8 and 14 were 5.90 (95% CI 4.94, 6.86) and 2.34 (95% CI 1.82, 2.87), respectively. The absolute numbers of VRE enumerated in fecal samples did not vary significantly between treatments (mixed effects negative binomial regression, Model two in *Supplementary file 1*; *Figure 2—figure supplement 1*).

The dramatic enrichment for daptomycin-resistant VR *E. faecium* in treated mice showed that subcutaneously-administered daptomycin produced GI tract concentrations high enough to select for resistance. To quantify fecal daptomycin concentrations, we analyzed fecal samples from a subset of daptomycin-treated mice by liquid chromatography-mass spectrometry (LC-MS) (*Figure 2C*). Samples from all time points tested (Days 2, 6, and 8) contained detectable daptomycin, and concentrations generally peaked at the end of treatment (Day 6). Higher daptomycin doses generally corresponded to higher fecal concentrations, but concentrations were highly variable and overlapped between treatments. While fecal VR *E. faecium* densities correlated poorly with the daptomycin dose administered (*Figure 2—figure supplement 1*), fecal VR *E. faecium* densities correlated with the amount of daptomycin recovered in feces (*Figure 2D*). These data confirmed that subcutaneously-administered daptomycin at our experimental doses generated a range of daptomycin concentrations in the GI tract that included inhibitory concentrations for the susceptible VR *E. faecium* strain.

We ran two additional experiments to further investigate the competitive dynamics between this susceptible and resistant pair (BL00239-1 and BL00239-1-R) in the presence and absence of daptomycin treatment. First, we tested whether susceptible bacteria competitively suppressed resistant bacteria in the GI tract. We inoculated mice with either a mixture of $10^8$ CFU susceptible + $10^3$ CFU-resistant VR *E. faecium*, or with a resistant-only inoculum at one of two inoculum sizes ($10^8$ CFU or $10^3$ CFU). Mice received 5 days of subcutaneous daptomycin injections at 100 mg/kg or control saline injections. Shedding of resistant and susceptible bacteria were quantified at time points throughout the experiment by plating (*Figure 3A*). In the absence of daptomycin treatment, the daptomycin-susceptible strain remained the most prevalent in mixed populations. When mixed populations were exposed to daptomycin, resistance increased to high frequency in three populations, and the population size fell dramatically in the remaining two populations. In mice inoculated with only $10^3$ CFU-resistant bacteria, the resistant clone was able to grow to high numbers with or without daptomycin. These data were consistent with the competitive suppression of the resistant strain by the susceptible strain in the absence of daptomycin treatment, and competitive release of the resistant strain during daptomycin treatment (*Day et al., 2015*; *Wargo et al., 2007*).

Next, we tested whether the frequency of daptomycin-resistant VR *E. faecium* would decrease over time in the absence of daptomycin treatment, potentially indicating that daptomycin-resistance comes at a fitness cost. We inoculated mice with a 1:5 mixture of daptomycin-resistant and susceptible VR *E. faecium*. Mice received no daptomycin treatment. After 14 days, the proportion of resistant bacteria had declined (one sample t-test, t = −22.42, df = 9, p<0.01, Cohen's d = 7.09), consistent with a competitive disadvantage (fitness cost) to the daptomycin-resistance mutation (*Figure 3B*). Control mice inoculated with only resistant or only susceptible bacteria did not have significantly different proportions of resistance between days 0 and 14 (resistant: t = −3.73, df = 2, p=0.06; susceptible: t = 1, df = 2, p=0.42).

## In vitro characterization of cholestyramine as a potential adjuvant

If an orally-administered adjuvant could reduce daptomycin activity in the GI tract, this could prevent the emergence of daptomycin-resistant *E. faecium* in the gut, potentially reducing transmission of resistant bacteria without impacting the effectiveness of intravenous daptomycin therapy. We identified cholestyramine, an FDA-approved bile-acid sequestrant, as a potential adjuvant for daptomycin therapy. Cholestyramine is a high-molecular weight anion exchange resin that binds with bile acids, forming an insoluble complex that is excreted in the feces (*Jacobson et al., 2007*). Cholestyramine is known to interact with a number of co-administered drugs through the same mechanism, reducing their bioactivity (*Jacobson et al., 2007*). We hypothesized that cholestyramine would bind daptomycin based on their chemical structures.

In vitro tests were consistent with cholestyramine binding daptomycin. Daptomycin solutions were incubated with cholestyramine, and then the cholestyramine was removed by centrifugation. The resulting supernatants were analyzed for changes in daptomycin concentration and activity. Daptomycin concentrations can be measured directly by ultraviolet (UV) absorbance at 364 nm (*Figure 4A*). Daptomycin concentrations were reduced in supernatants after incubation with cholestyramine in a dose-dependent manner (*Figure 4B*). Additionally, daptomycin solutions incubated with cholestyramine had reduced antibiotic activity against *E. faecium* in broth microdilution (*Figure 4C*). Together, these results were consistent with cholestyramine removing daptomycin from solution, supporting cholestyramine as a candidate adjuvant for daptomycin therapy.

## Adjunctive cholestyramine therapy prevents emergence of daptomycin-resistance

We conducted four experiments to test whether adjunctive therapy with cholestyramine could prevent the emergence of daptomycin-resistant VR *E. faecium* in the mouse GI tract. In each experiment, mice were orally inoculated with a 1:20 mixture of daptomycin-resistant and susceptible VR *E. faecium* and then treated with subcutaneous daptomycin injections for 5 days. Densities of total VR *E. faecium* and daptomycin-resistant VR *E. faecium* were determined by plating (*Figure 5*, *Figure 5—figure supplements 1–5*). The experiments tested the evolutionary impact of oral cholestyramine in different mouse strains, with different VR *E. faecium* strains, and with different timing of cholestyramine administration. The design of the four experiments was as follows: (A) Swiss Webster mice with *E. faecium* strains BL00239-1 and BL00239-1-R, with cholestyramine started one day before daptomycin (*Figure 5—figure supplement 1*), (B) C57BL/6 mice with *E. faecium* strains BL00239-1 and BL00239-1-R, with cholestyramine started one day before daptomycin (*Figure 5—figure supplement 2*), (C) Swiss Webster mice with *E. faecium* strains PR00708-14 and PR00708-14-R, with cholestyramine started one day before daptomycin (*Figure 5—figure supplement 3*), and (D) Swiss Webster mice with *E. faecium* strains BL00239-1 and BL00239-1-R, with cholestyramine started the same day as daptomycin (*Figure 5—figure supplement 4*). Data from these experiments were analyzed together, with a block effect included in the models. Because bacterial densities were found not to correlate to daptomycin dose (*Figure 2—figure supplement 1*), all daptomycin doses were combined into a single group for analysis. *Figure 5—figure supplements 1–4* show these data broken down by experiment and by daptomycin dose. For daptomycin-treated mice shedding detectable levels of VR *E. faecium* by our plating assay (at least 20 CFU/10 mg feces at a given time point), the cholestyramine-supplemented diet reduced the proportion of daptomycin-resistant VR *E. faecium* (mixed effects binomial regression, p<0.01, Model three in *Supplementary file 1*). The effect size (Cohen's d) of cholestyramine diet on the proportion of resistant bacteria in daptomycin-treated mice was 0.41 (95% CI 0.01, 0.82) at Day 2, 0.56 (95% CI 0.13, 1.00) at Day 4, 0.90 (95% CI 0.45, 1.35) at Day 6, 1.08 (95% CI 0.63, 1.53) at Day 8, and 0.63 (95% CI 0.21, 1.05) at Day 14. In addition, to more accurately determine resistance proportions for Days 8 and 14, we plated an estimated 200 CFU from each sample in triplicate on plates with and without daptomycin (*Figure 5—figure supplement 5*). This second assay confirmed that cholestyramine reduced the proportion of resistant VR *E. faecium* at these time points (mixed effects binomial regression, p<0.01, Model four in *Supplementary file 1*; Cohen's d 1.25 (95% CI 0.97, 1.53) at Day 8, 0.73 (95% CI 0.48, 0.98) at Day 14).

We also quantified absolute densities of daptomycin-resistant and susceptible VR *E. faecium* over time by plating samples from Days 0, 1, 2, 4, 6, 8, and 14 (*Figure 5*). These data showed that the cholestyramine-supplemented diet reduced fecal shedding of daptomycin-resistant VR *E. faecium* in daptomycin-treated mice (Antibiotic*Diet*Day interaction p<0.01, Model five in *Supplementary file 1*). The effect size was greatest in the days after daptomycin treatment. The effect size (Cohen's d) of cholestyramine diet on shedding of resistant bacteria in daptomycin-treated mice was 0.43 (95% CI 0.02, 0.83) at Day 2, 0.08 (95% CI −0.32, 0.48) at Day 4, 0.12 (95% CI −0.28, 0.51) at Day 6, 0.36 (95% CI −0.04, 0.76) at Day 8, and 0.57 (95% CI 0.16, 0.98) at Day 14.

Total VR *E. faecium* shedding was also influenced by the addition of cholestyramine (Antibiotic*Diet*Day interaction p<0.01, Model six in *Supplementary file 1*). Here the effect size is greatest during daptomycin treatment. The effect size (Cohen's d) of cholestyramine diet on total shedding in daptomycin-treated mice was 0.84 (95% CI 0.43, 1.23) at Day 2, 0.36 (95% CI −0.04, 0.76) at Day 4, 0.31 (95% CI −0.09, 0.71) at Day 6, 0.04 (95% CI −0.36, 0.43) at Day 8, and 0.40 (95% CI 0.00,

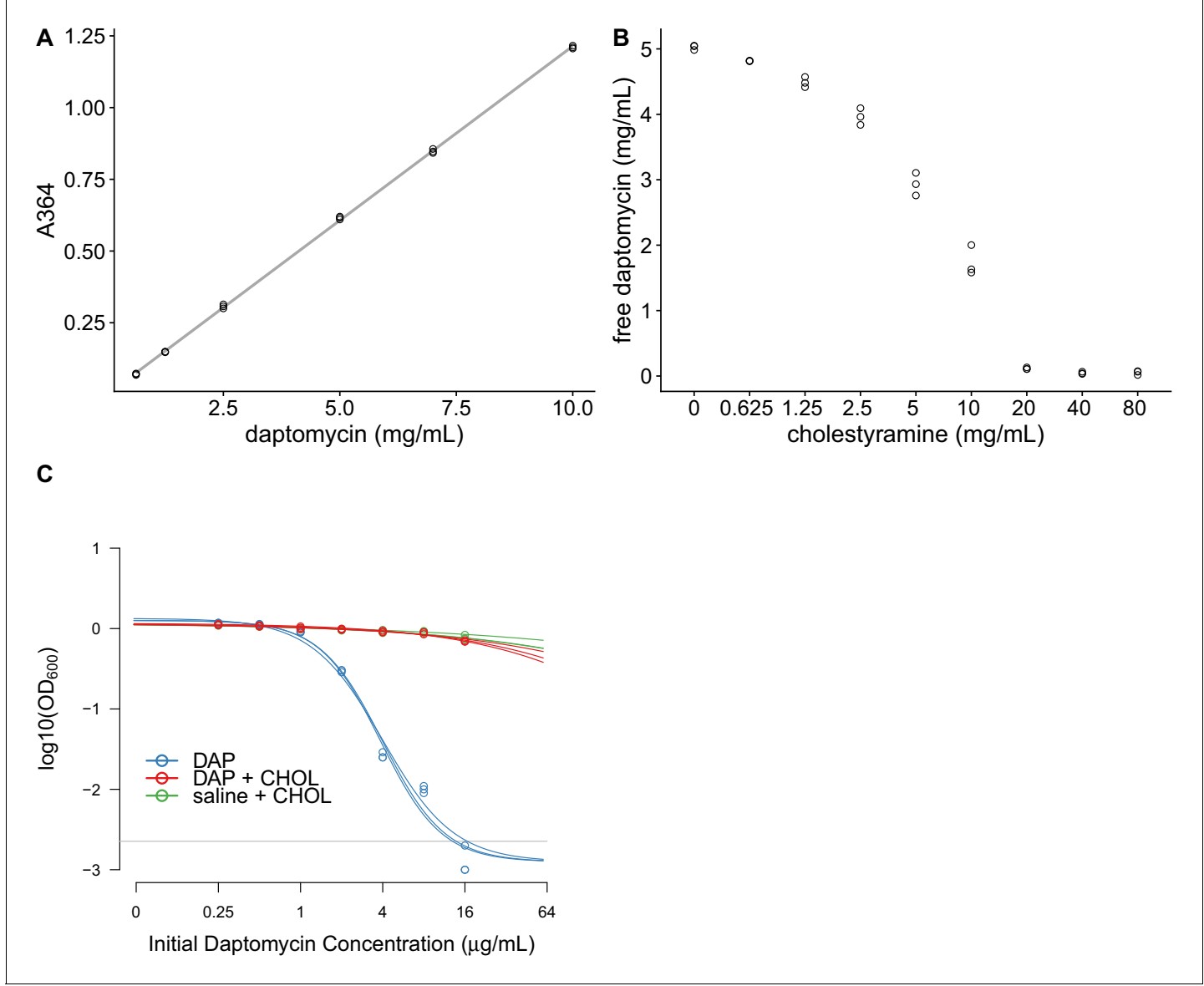

**Figure 4.** Cholestyramine captures daptomycin in vitro. (**A**) Calibration curve (best fit linear regression) showing that daptomycin concentration can be measured by UV absorption at 364 nm (N = 3 per concentration). (**B**) Daptomycin concentration was reduced in solutions treated with cholestyramine (N = 3 per concentration). (**C**) Daptomycin solutions treated with cholestyramine had reduced biological activity against VRE in broth microdilutions (N = 3 per antibiotic treatment). Bacterial densities ($OD_{600}$) following growth in the presence of daptomycin (DAP), daptomycin solution treated with cholestyramine (DAP + CHOL), or saline solution treated with cholestyramine (saline + CHOL) are shown. Concentrations are shown as the initial concentration of daptomycin in solution prior to cholestyramine treatment. Saline controls were constant across all listed concentrations. Horizontal line shows detection threshold.

The online version of this article includes the following source data for figure 4:

**Source data 1.** Calibration curve data (*Figure 4A*).

**Source data 2.** Daptomycin concentration data (*Figure 4B*).

**Source data 3.** Broth microdilution data, $OD_{600}$ readings from 96-well plate (*Figure 4C*).

0.80) at Day 14. If we consider only the control treatments, where there is no possibility of cholestyramine protecting against daptomycin killing, the addition of cholestyramine to the diet does not significantly influence total VR *E. faecium* counts alone (p=0.63, Model seven in *Supplementary file 1*) or in combination with day (p=0.18).

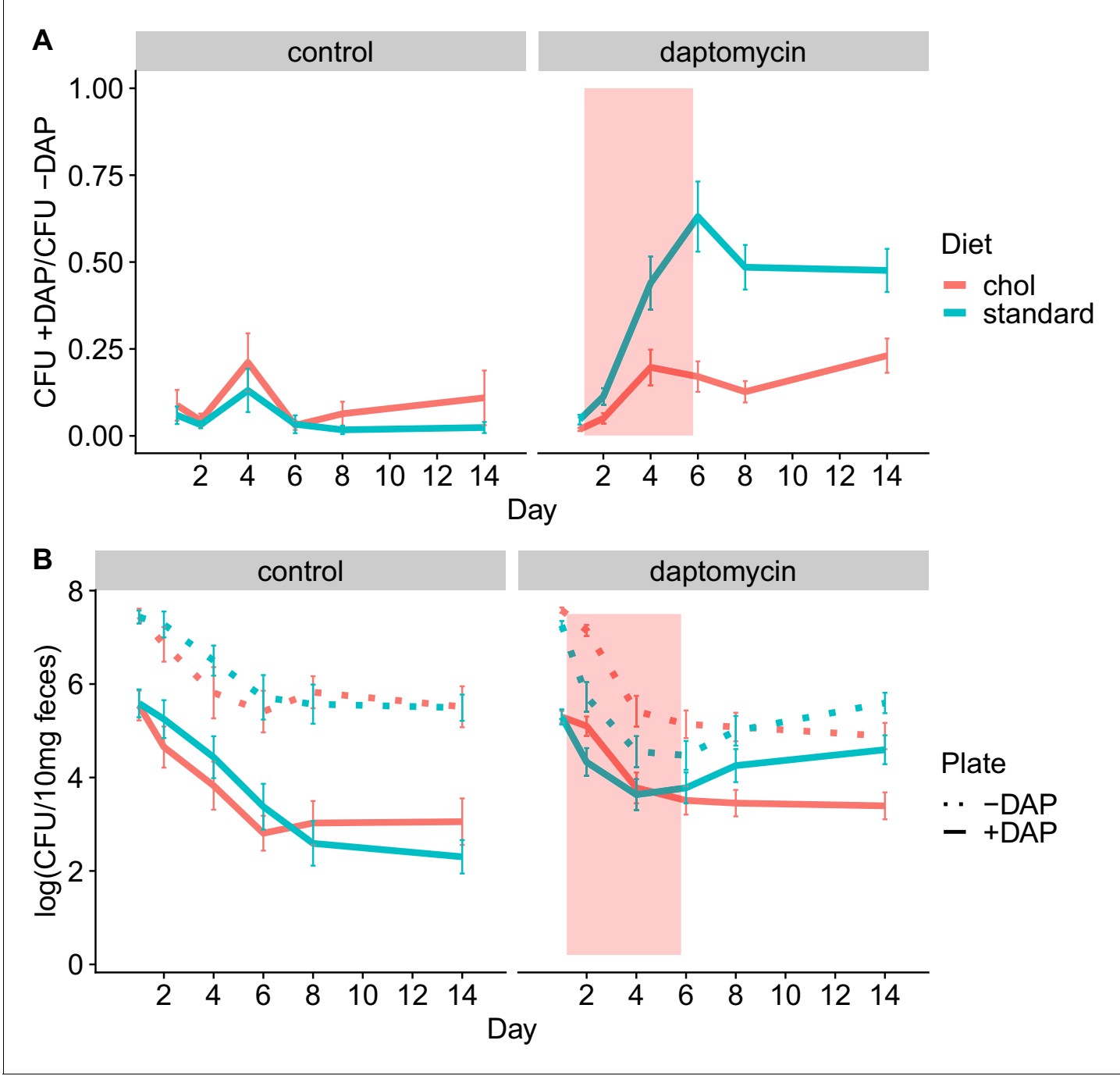

**Figure 5.** Adjunctive cholestyramine reduces enrichment of daptomycin-resistance in the GI tract. (A) The proportion of fecal VR *E. faecium* that were daptomycin-resistant over time in mice. Proportions were determined by plating on agar with daptomycin (+DAP) and without daptomycin (-DAP). Data shown were combined from four experiments (for each diet N = 20 controls and N = 50 daptomycin-treated, mean + SEM shown). Proportions were not determined for samples with <20 CFU VR *E. faecium* per 10 mg feces, as these densities were below the limit of detection for this plating assay, and these samples were not included in Panel A. The pink shaded region indicates days of daptomycin therapy. (B) Total VR *E. faecium* densities corresponding to data shown in Panel A (for each diet N = 20 controls and N = 50 daptomycin-treated, mean + SEM shown). Dotted line shows total density (-DAP) and solid line shows the density of daptomycin-resistant VR *E. faecium* (+DAP). All samples, including those with low densities, were included in Panel B.

The online version of this article includes the following source data and figure supplement(s) for figure 5:

**Source data 1.** VR *E. faecium* fecal density data.

**Figure supplement 1.** Fecal shedding of VR *E. faecium* for Experiment A (strain BL00239-1, Swiss Webster mice, cholestyramine started one day prior to first daptomycin dose).

*Figure 5 continued on next page*

*Figure 5 continued*

**Figure supplement 2.** Fecal shedding of VR *E.* faecium for Experiment B (strain BL00239-1, C57BL/6 mice, cholestyramine started one day prior to first daptomycin dose).

**Figure supplement 3.** Fecal shedding of VR *E. faecium* for Experiment C (strain PR00708-14, Swiss Webster mice, cholestyramine started one day prior to first daptomycin dose).

**Figure supplement 4.** Fecal shedding of VR *E. faecium* for Experiment D (strain BL00239-1, Swiss Webster mice, cholestyramine started same day as first daptomycin dose).

**Figure supplement 5.** Adjunctive cholestyramine prevents emergence of daptomycin-resistance in GI tract.

## Discussion

Here we have shown proof of concept for an adjunctive therapy approach to prevent the emergence of daptomycin-resistant *E. faecium* in the GI tract during daptomycin therapy. Ideally, this approach would allow clinicians to treat bloodstream infections with intravenous daptomycin without fueling the hospital transmission of multidrug-resistant bacteria. This would be a novel approach because the desired outcome is reduced resistance evolution and reduced transmission of resistant pathogens. Colonization with VR *E. faecium* is a risk factor for future infection (*Alevizakos et al., 2017*; *Olivier et al., 2008*), presumably because GI tract populations are sources for infections. By maintaining daptomycin sensitivity GI tract populations, this strategy could reduce the risk of resistant infection for colonized patients, as well as reducing transmission of resistance between patients. With some optimization, this adjunctive approach could be implemented in hospitals. Cholestyramine is an inexpensive, FDA-approved drug with few side effects (*Beckett and Wilhite, 2015*; *Jacobson et al., 2007*). Cholestyramine has been used clinically for over 50 years, including in hospital patient populations that would be targets for adjunctive cholestyramine therapy (*Ballantyne, 2009*). Cholestyramine does have potential to interfere with other orally administered drugs, which could be a risk considered on a patient by patient basis (*Scaldaferri et al., 2013*). This drug–drug interaction potential is managed in practice by administering oral drugs an hour before or 4 to 6 hr after cholestyramine administration. Because the effects and risks of cholestyramine are well understood, testing cholestyramine as an adjuvant with daptomycin in human trials is appealingly low risk to patient health while offering large potential gains to hospital infection control.

While cholestyramine has the potential to be repurposed in hospitals, this study also revealed potential limitations of cholestyramine therapy and areas requiring further study prior to clinical trials. Cholestyramine therapy was most effective in a narrow range of conditions (intermediate daptomycin doses, cholestyramine started prior to daptomycin therapy) (*Figure 5—figure supplements 1–5*), and the daptomycin dose at which resistance was most likely to be enriched varied between *E. faecium* strains (*Figure 5—figure supplements 1–5*). The effect of cholestyramine was also variable among mice (*Figure 5—figure supplements 1–5*), possibly as a result of variable daptomycin concentrations in the intestines (*Figure 2C–D*). This is potentially a major limitation for translating this therapy, as daptomycin pharmacokinetics and pharmacodynamics also vary among human patients. Optimizing cholestyramine therapy will require detailed data on intestinal daptomycin pharmacokinetics, including temporal variation over the course of treatment and variation among individuals. From these data, one could estimate a maximum target for daptomycin capture, and optimize cholestyramine dosing accordingly. This therapeutic approach has the potential to be expanded to other pathogen-drug combinations, but there is likely no one size fits all adjunctive therapy; optimization may be required for each pathogen-drug combination.

In addition to binding antibiotics, cholestyramine sequesters bile acids in the intestine, which alters the GI tract environment. This means that cholestyramine treatment could indirectly affect bacteria in the GI tract through mechanisms other than direct interactions with drugs. For example, exposure to bile acids has implications for *E. faecium* phenotypes. Secondary bile acids trigger a morphotype switch in *Enterococcus* that facilitates intestinal colonization and biofilm formation (*McKenney et al., 2019*). In our experiments, we did not observe differences in colonization efficiency or VR *E. faecium* counts in mice treated with cholestyramine alone. In mouse models, cholestyramine has also been shown to reduce bile acid-mediated resistance to *Clostridium difficile* infection, which is a possible risk associated with cholestyramine treatment (*Buffie et al., 2015*).

Here we tested cholestyramine as an adjuvant to reduce selection for antibiotic resistance in the GI tract, but other adjuvants could be used or designed for the same purpose. At least two other drugs that reduce antibiotic activity in the GI tract are currently in development. DAV-132, a formulation of activated charcoal encased in zinc-pectinate beads, was shown to site-specifically bind antimicrobials in the gut in a stage I clinical trial (*de Gunzburg et al., 2018*; *de Gunzburg et al., 2015*). DAV-132 has been shown to reduce fecal concentrations of antibiotics by 99% without affecting plasma concentrations (*Burdet et al., 2017*; *de Gunzburg et al., 2018*; *de Gunzburg et al., 2015*; *Khoder et al., 2010*). Activated charcoal could adsorb antibiotics in the gut, and could be an effective adjuvant for a broad range of antibiotic classes. Adjuvants with specific activity against particular antibiotics could also be developed. This has been successfully demonstrated with orally-administered β-lactamases given with intravenous β-lactam antibiotics. β-lactamases enzymatically inactivate β-lactams. Under the name SYN-004, this β-lactamase treatment advanced to clinical trials in human subjects (*Kaleko et al., 2016*; *Kokai-Kun et al., 2017*; *Pitout, 2009*; *Tarkkanen et al., 2009*). Data from clinical trials show the drugs successfully inactivate β-lactams in the digestive tract without adversely affecting levels of antibiotic in plasma (*Kaleko et al., 2016*; *Kokai-Kun et al., 2017*; *Pitout, 2009*; *Tarkkanen et al., 2009*). These drugs have been developed with the goal of preventing infection with *C. difficile* after antibiotic therapy, but they could likely also prevent the emergence of antibiotic resistance in the GI tract. The orally-administered β-lactamase SYN-006 mitigated the enrichment of genes associated with antibiotic resistance in the microbiomes of pigs treated with a carbapenem antibiotic (*Connelly et al., 2019*). So far as we are aware, there is no direct experimental evidence analogous to ours that those drugs can prevent resistance emergence in colonizing opportunistic pathogens, but it seems likely they could.

Adjunctive therapies like the one proposed here could help manage resistance evolution in other important pathogens listed by Centers for Disease Control and Prevention (CDC) and World Health Organization (WHO) as major threats (*CDC, 2019*; *WHO, 2017*). Like VRE, many opportunistic pathogens experience substantial antibiotic exposure when they are not the targets of treatment. Opportunistic pathogens like *Klebsiella pneumoniae*, *Escherichia coli*, *Staphylococcus aureus*, and *Enterobacter cloacae* colonize the gut asymptomatically, where they can be unintentionally exposed to antibiotics (*Morley et al., 2019*). According to one estimate, over 90% of the total antimicrobial exposure experienced by *K. pneumoniae* occurs when *K. pneumoniae* was not the target of treatment (*Tedijanto et al., 2018*). For *H. influenzae*, *E. coli*, and *Staphylococcus aureus* over 80% of total exposure to antibiotics was estimated to occur when the bacteria were bystanders (*Tedijanto et al., 2018*). These colonizing populations are sources for infections, so selection for resistance in bystander populations can contribute to rising rates of resistant infections (*Morley et al., 2019*). Adjunctive strategies that allow intravenous antibiotics to reach target sites while reducing off-target exposure could help stem the spread of resistant pathogens listed by the CDC as urgent or serious threats (*CDC, 2019*), including carbapenem-resistant Enterobacteriaceae (CRE), extended-spectrum beta-lactamase producing Enterobacteriaceae, vancomycin-resistant *Enterococcus* (VRE), methicillin-resistant *Staphylococcus aureus* (MRSA), and drug-resistant *Staphylococcus aureus*. The strategy presented here focuses on inactivating antibiotic in the gastrointestinal tract, the likely source of the bulk of the antimicrobial resistance in pathogens listed as top threats by the CDC, but similar strategies could be developed to shield microbiota at other sites, such as the skin and respiratory tract.

While adjunctive therapies have potential to be used for a variety of pathogens, the genetic mechanism of resistance will influence the resulting evolutionary dynamics. The cholestyramine therapy described here prevents enrichment of resistant clones within a bacterial population consisting of both sensitive and resistant bacteria. Daptomycin-resistance in *E. faecium* emerges through point mutations in the genome, so patients colonized with daptomycin-susceptible *E. faecium* likely harbor low-frequency resistant clones due to spontaneous mutation. Other forms of resistance, like VanA-type vancomycin resistance in *E. faecium*, emerge through horizontal gene transfer rather than point mutations (*Depardieu and Courvalin, 2017*). For resistance to emerge in patient colonized with vancomycin-susceptible *E. faecium*, a horizontal transfer event would have to occur, or a new resistant colonizer would have to be introduced. If these events are rarer than spontaneous mutation, patients are less likely to be colonized with a mixture of vancomycin sensitive and resistant bacteria than with daptomycin-resistance, as studied here. However, gains and losses of transmissible resistance elements have been observed in the gut microbiomes of patients (*Kinnear et al., 2020*), and altering

antibiotic pressure in the gut could reduce selection for resistant bacteria, even when that resistance is encoded by horizontally transferred genes.

## Materials and methods

### Key resources table

| Reagent type (species) or resource | Designation | Source or reference | Identifiers | Additional information |
|---|---|---|---|---|
| Strain, strain background (mouse, female) | Swiss Webster (CFW) | Charles River | | |
| Strain, strain background (mouse, female) | C57BL/6 | Charles River | | |
| Chemical compound, drug | cholestyramine | Sigma-Aldrich | cat #: C4650 | |
| Strain, strain background (E. faecium) | BL00239-1 | This paper | | clinical bloodstream isolate, R. Woods lab, University of Michigan |
| Strain, strain background (E. faecium) | PR00708-14 | This paper | | clinical perirectal swab isolate, R. Woods lab, University of Michigan |
| Strain, strain background (E. faecium) | BL00239-1-R | This paper | | derived from evolution in mouse gut |
| Strain, strain background (E. faecium) | PR00708-14-R | This paper | | derived from evolution in mouse gut |

### Mice and bacterial strains

Unless otherwise specified, mice in all experiments were female Swiss Webster. In one experiment, inbred female C57BL/6 mice were used to ensure the results were not specific to one mouse strain. Mice were fed a standard diet (5001 Laboratory Rodent Diet) or a standard diet supplemented with 2% w/w cholestyramine resin. All mice were housed individually during experiments.

Daptomycin-susceptible VR *E. faecium* strains were isolated from different patients at the University of Michigan Hospital. Strain BL00239-1 was isolated from a bloodstream infection, and strain PR00708-14 was isolated from a perirectal swab. Additional strains were isolated during these experiments from mouse fecal samples (including BL00239-1-R and PR00708-14-R). These strains were isolated by streaking on agar plates for two rounds of clonal purification.

### Daptomycin treatment experiments

All mice were pretreated with ampicillin (0.5 g/L in drinking water) for 7 days before *E. faecium* inoculation. Ampicillin disrupts the natural gut flora and facilitates *Enterococcus* colonization (*McKenney et al., 2019*). Sample sizes for mouse experiments were chosen based on previous experience with similar experiments. Mice that were co-housed during ampicillin pre-treatment were evenly allocated among experimental treatment groups. *E. faecium* strains were plated from glycerol stocks and then grown overnight in liquid culture in Brain Heart Infusion broth. Mice were inoculated via oral gavage with $10^8$ CFU *E. faecium* suspended in saline. *E. faecium* inoculum counts were confirmed by plating. Following *E. faecium* inoculation, mice were split into individual cages with untreated water and any experimental diets. Daptomycin doses were administered daily starting one day post-inoculation via subcutaneous injection or oral gavage. Daptomycin doses were based on an average mouse weight for each experiment. In some experiments, a cholestyramine-supplemented diet (2% w/w) was provided to mice starting one day prior to the first daptomycin dose (Experiments A-C) or starting the same day as the first daptomycin dose (Experiment D). Once started, mice were maintained on the cholestyramine diet for the duration of the experiment. For stool collection, mice were placed in clean plastic cups, and fresh stool was collected using a sterile toothpick. Stool

samples were suspended in PBS (25 uL PBS/mg stool) and frozen with glycerol at −80°C for subsequent analysis.

## In vitro tests of daptomycin interaction with cholestyramine

For measurements of UV absorbance, solutions of 5 mg/mL daptomycin in phosphate-buffered saline (PBS) were combined with various concentrations of cholestyramine. These mixtures were vortexed for 30 s, then allowed to incubate for 5 min at room temperature. Following incubation, cholestyramine was removed by centrifugation. Supernatants were analyzed for absorbance at 364 nm on a NanoVue Plus Spectrophotometer. A calibration curve was used to determine daptomycin concentrations from A364 values.

For tests of daptomycin bioactivity, solutions of 1 mg/mL daptomycin were incubated with or without 12 mg/mL cholestyramine for 45 min at 37°C with shaking (N = 3 per treatment). The cholestyramine was removed by centrifugation, and the supernatant was used in broth microdilutions with *E. faecium*. Saline solution incubated with cholestyramine run as a control had no effect on cell growth.

## Analysis of VRE in stool samples

VR *E. faecium* were enumerated by plating diluted fecal suspensions on selective plates (Enterococcosel agar supplemented with 16 µg/mL vancomycin). Plates were incubated at 35°C for 40–48 hr, and colonies were counted. To quantify the proportion of these bacteria that were daptomycin-resistant, fecal suspensions were plated on calcium-supplemented Enterococcosel plates with 16 µg/mL vancomycin and 10 µg/mL daptomycin. Plates were incubated at 35°C for 40–48 hr, and colonies were counted. Serially-diluted fecal suspensions were each plated once on plates without daptomycin and once on plates containing daptomycin to estimate the proportion of daptomycin-resistant bacteria. An additional plating assay was performed to more accurately determine proportions of resistant bacteria for Days 8 and 14 in each experiment. For this assay, known sample densities were used to plate an estimated 200 CFU on plates with and without daptomycin in triplicate. This assay only included samples with high enough initial bacterial density to generate 200 CFU on each plate ($3 \times 10^3$ CFU/10 mg in initial sample). Counters were not blinded to treatment groups.

In some experiments, *E. faecium* clones were isolated from fecal samples and analyzed by broth microdilution. Clones were purified by streaking twice on Enterococcosel agar with 16 µg/mL vancomycin, and were then stored in glycerol stocks at −80°C. Broth microdilutions were performed according to Clinical and Laboratory Standards Institute (CLSI) guidelines (*CLSI, 2017*). After incubation, cell densities were measured by OD600 absorbance in a plate reader. OD values were fitted to a Hill function curve to determine the computed MIC ($MIC_c$) as described previously (*Kinnear et al., 2020*).

## Genome sequencing

Whole genomic DNA preparations were submitted to the University of Michigan sequencing core for Illumina library preparation and paired end sequencing with Illumina NovaSeq 6000 (isolates PR00708-14 and PR00708-14-R) or Illumina HiSeq (isolates BL00239-1 and BL00239-1-R). Long read data was additionally generated for strain BL00239-1 using the Oxford Nanopore MinION. The nanopore library were prepared using the Nanopore Ligation Sequencing Kit (SQK-LSK109). Quality control of sequencing reads was performed using Trimmomatic (*Bolger et al., 2014*). De novo genome assembly was performed using SPAdes (*Bankevich et al., 2012*) and genomes were annotated using Prokka (*Seemann, 2014*). Trimmed reads from resistant isolates were mapped against corresponding susceptible reference genomes using Burrows-Wheeler Aligner (BWA) (*Li and Durbin, 2009*), and candidate variants were identified with The Genome Analysis Toolkit (GATK) (*McKenna et al., 2010*) or breseq (*Deatherage and Barrick, 2014*). Reads from the reference sample were aligned to the reference genome (aligned to self) to generate a list of background variants; these background variants were filtered out during variant calling. Remaining candidate variants were screened by visual inspection of alignments in Integrative Genomics Viewer (IGV) (*Thorvaldsdottir et al., 2013*).

### Analysis of daptomycin concentrations

Fecal daptomycin concentrations were measured via LC-MS at the University of Michigan Pharmacokinetics Core. A labeled daptomycin-d5 internal standard was used to generate calibration curves.

### Statistical analysis

Statistical analyses were run in R v1.2.1335 (*Brooks et al., 2017*) using the packages 'nlme' (*Pinheiro et al., 2016*) and 'glmmTMB' (*Brooks et al., 2017*). To analyze proportions of resistant bacteria, samples were plated on agar with and without daptomycin, resulting in a count of resistant bacteria and a count of total bacteria. Due to sampling, these proportions were not bounded by zero and one, so proportion data were normalized by dividing each value by the maximum value in the data set. Proportions of resistant bacteria were analyzed using mixed binomial regression models. Absolute VRE densities were analyzed using mixed models with an autoregressive error structure as previously described (*Pollitt et al., 2012*). Full model structures and output are shown in *Supplementary file 1*.

## Acknowledgements

We thank Kevin Tracy for assistance in generating the nanopore sequencing and analysis pipeline. We are grateful to Vaughn Cooper, Luke McNally, and an anonymous reviewer for comments that greatly improved this manuscript.

## Additional information

### Funding

| Funder | Grant reference number | Author |
| --- | --- | --- |
| Eberly College of Science, Penn State | | Andrew F Read |
| Eberly Family Trust | | Andrew F Read |
| NIH | R01 AI143852 | Robert J Woods |

The funders had no role in study design, data collection and interpretation, or the decision to submit the work for publication.

### Author contributions

Valerie J Morley, Conceptualization, Data curation, Formal analysis, Investigation, Visualization, Methodology, Writing - original draft, Project administration, Writing - review and editing; Clare L Kinnear, Conceptualization, Resources, Methodology, Writing - review and editing; Derek G Sim, Grace A Usher, Investigation, Methodology, Writing - review and editing; Samantha N Olson, Lindsey M Jackson, Investigation, Writing - review and editing; Elsa Hansen, Software, Writing - review and editing; Scott A Showalter, Methodology, Writing - review and editing; Manjunath P Pai, Formal analysis, Investigation, Methodology, Writing - review and editing; Robert J Woods, Conceptualization, Resources, Formal analysis, Supervision, Methodology, Project administration, Writing - review and editing; Andrew F Read, Conceptualization, Supervision, Funding acquisition, Methodology, Project administration, Writing - review and editing

### Author ORCIDs

Valerie J Morley (ID) https://orcid.org/0000-0001-6805-7562
Clare L Kinnear (ID) http://orcid.org/0000-0002-4126-7540
Lindsey M Jackson (ID) http://orcid.org/0000-0002-0559-9433
Grace A Usher (ID) http://orcid.org/0000-0002-8303-9992
Andrew F Read (ID) https://orcid.org/0000-0001-7604-7903

## Ethics

Animal experimentation: All of the animals were handled according to approved institutional animal care and use committee (IACUC) protocols of the Pennsylvania State University (approved IACUC protocol #47581).

## Decision letter and Author response

Decision letter https://doi.org/10.7554/eLife.58147.sa1
Author response https://doi.org/10.7554/eLife.58147.sa2

# Additional files

## Supplementary files

- Supplementary file 1. Statistical models.
- Transparent reporting form

## Data availability

Raw experimental data are available in Dryad (doi:10.5061/dryad.qrfj6q5c2) and sequence data have been submitted to NCBI SRA (BioProject PRJNA629735).

The following datasets were generated:

| Author(s) | Year | Dataset title | Dataset URL | Database and Identifier |
|---|---|---|---|---|
| Morley VJ, Kinnear CI, Sim DG, Olson SN, Jackson Lm, Hansen E, Usher GA, Showalter SA, Pai MP, Woods RJ, Read AF | 2020 | An adjunctive therapy administered with an antibiotic prevents enrichment of antibiotic-resistant clones of a colonizing opportunistic pathogen | https://doi.org/10.5061/dryad.qrfj6q5c2 | Dryad Digital Repository, 10.5061/dryad.qrfj6q5c2 |
| Morley VJ, Kinnear CI, Sim DG, Olson SN, Jackson Lm, Hansen E, Usher GA, Showalter SA, Pai MP, Woods RJ, Read AF | 2020 | An adjunctive therapy approach prevents antibiotic resistance emergence in opportunistic pathogens colonizing the gut | https://www.ncbi.nlm.nih.gov/bioproject/629735 | NCBI BioProject, PRJNA629735 |

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
