## [Decision Letter]

**Acceptance summary:**

This is a creative study that tests the theory that adjuvants can suppress resistance evolution in vivo. Specifically, the authors evaluate whether the adjuvant cholestyramine can suppress the evolution of resistance to daptomycin, an important part of the current arsenal, in a mouse VR *E. faecium* gut colonization model. The motivation is clear because daptomycin is given to treat sepsis, where resistance evolution would be a dead-end, but often also selects for resistance in the gut where it is transmissible.

**Decision letter after peer review:**

Thank you for submitting your article "An adjunctive therapy approach prevents antibiotic resistance emergence in opportunistic pathogens colonizing the gut" for consideration by *eLife*. Your article has been reviewed by three peer reviewers, including Vaughn S Cooper as the Reviewing Editor and Reviewer #1, and the evaluation has been overseen by Dominique Soldati-Favre as the Senior Editor. The following individual involved in review of your submission has agreed to reveal their identity: Luke McNally (Reviewer #3).

The reviewers have discussed the reviews with one another and the Reviewing Editor has drafted this decision to help you prepare a revised submission.

Summary:

This is an elegant study designed to test if the adjuvant cholestyramine can suppress the evolution of resistance to daptomycin, an important part of the current arsenal, in a mouse VR *E. faecium* gut colonization model. The motivation is clear because daptomycin is given to treat sepsis, where resistance evolution would be a dead-end, but often also selects for resistance in the gut where it is transmissible. Overall, the premise is strong and the experimental design well described.

1) All reviewers found that the manuscript overstates the generalizability of the findings and the results are more inconsistent than acknowledged. The preventative effect they measured was limited to a narrow range of conditions (Figure 6) and the effect was inconsistent (Supplementary figure 2). First, in the absence of cholestyramine, a relative enrichment of the resistant strain was observed in 4 of the 6 dose-strain combinations (Figure 6A and C), but with an inconsistent dose-response (for Strain PR00708-14, in the absence of cholestyramine, relative enrichment was observed at a daptomycin dose of 100 mg/kg and 200 mg/kg, but not 150 mg/kg). There is significant variability in the pharmacokinetics and pharmacodynamics of antibiotics in humans too, so the narrow dose-effect range observed in this study may be limiting. Also, it may differ substantially for every bug-drug combination. The etiology of this narrow effect window is not known and was not investigated. The authors should discuss this finding as a possible major limitation of the model and potentially to the future application of this overall strategy.

2) Additionally, when cholestyramine was administered on the same day as daptomycin, no effect, or perhaps even the opposite effect, was observed (Supplementary figure 2). The data in Supplementary figure 2 is not discussed in the manuscript and is cited in the Materials and methods section with the sentence, "Preliminary experiments were preformed to determine the timing of experimental conditions (Supplementary figure 2)". However, the data presented in Supplementary figure 2 contradicts the main conclusion of the manuscript. The authors should provide further clarity because if one considers all of the data provided, the reproducibility of the major finding described in the manuscript might be significantly diminished.

3) The data in Supplementary figure 2 should be combined with the data in Figure 6, so that the reader can assess all of the data in aggregate. In fact, given bacterial densities were found to not correlate with daptomycin dose (Figure 2—figure supplement 1), it would be reasonable, for each bacterial strain, to combine all doses of daptomycin into a single group. This aggregate data should be presented in the main text and presented with a statistical evaluation of the effect size (comparing “standardl” vs. “chol”) at each time point. This evaluation of the data would be more logical and greatly simplify its presentation. The parsed-out data of individual daptomycin doses could still be presented in the supplemental data section.

4) The genome sequencing results are unacceptably vague and the requisite data do not appear to be available. You state "We confirmed that the isolated clone (PR00708-14-R) had an increased daptomycin MICc relative to PR00708-14 by broth microdilution (MICc = 12.0). Genome sequencing revealed that PR00708-14 and PR00708-14-R differed by several mutations in hypothetical proteins and noncoding regions, which to our knowledge have not previously been associated with daptomycin-resistance.” Please be specific. Further, this one limited example does not show that daptomycin-resistance evolved in vivo consistently. One could argue that the populations were uncontrolled at some concentrations and grew steadily to higher levels because of some physiological tolerance. Although this is unlikely, it would not be hard to pick a few more clones and confirm in fact that they evolved heritable resistance and identify the causative mutations, which you would report.

5) The distinction between preventing de novo emergence of a resistant pathogen and enrichment of a resistant colonizer is important and mechanistically distinct. For mechanisms of antibiotic resistance (such as daptomycin), where point mutations in the genome are causative, resistant clones within susceptible populations are likely to be present at very low frequencies simply due to spontaneous mutation. In this case, this approach does have promise because, in theory, all patients receiving daptomycin would be “at-risk” of developing de novo emergence of daptomycin-resistance. However, for mechanisms of antibiotic resistance that rely on horizontal gene transfer (HGT) or strains with intrinsic resistance, resistant clones do not arise spontaneously from a susceptible population. Instead, a HGT event must take place, or the patient must become newly colonized with that resistant strain. These are relatively rare events. For example, vancomycin resistance in Enterococcus (VRE) is caused by a van gene cassette. This multi-gene cassette does not arise spontaneously. Bacteria get this by HGT and patients get this by being inoculated with VRE in the hospital and, presumably, VRE colonization is sustained in the setting of repeated vancomycin exposures. Thus, a strategy for preventing enrichment of VRE:VSE would only work for patients that are already colonized with VRE. Consider drawing this distinction in the Discussion.

6) There is insufficient detail of the statistical models used by the authors to assess their validity. The full structure of all reported mixed effects models should be specified. In addition, only p-values for the statistical tests are reported. Effect sizes and their error should also be reported. Full tables of all estimated parameters, including for random effects, from the mixed effects models should be reported.

7) For analysis of time-series of VRE it is stated that interpolated density values were used. It needs to be clarified why and how this interpolation was carried out, and whether this has inflated the authors' sample size.

8) It is simply stated by the authors that model selection was carried out based on AIC. If model selection is to be carried out tables of AIC values should be reported for all fitted models and details of the model selection process need to be given. However, I don't believe model selection is appropriate here. It is well known that model selection based on AIC leads to inflated type I error rates via cryptic multiple hypothesis testing (https://link.springer.com/article/10.1007/s00265-010-1038-5). Statistical results should probably simply be reported from the full model.

9) The authors use negative binomial mixed effects models to analyse their data on proportions of resistant bacteria. This appears an odd choice, as a negative binomial distribution is typically used to model abundances. For analysing proportions given from counts a binomial model would be more appropriate. The authors either need to clarify exactly why they have used a negative binomial distribution or re-run these as binomial models.

Revisions expected in follow-up work:

1) In these experiments, the resistant clones persisted to day 14, so, further studies are needed to test whether this strategy can interrupt stable colonization, which may be a more relevant end-point than preventing enrichment. However, it should be mentioned in the Discussion that some studies have shown *E. faecium* outgrowth in the gut precedes clinical infections, as this is pertinent to this strategy and gives this study more merit and relevance.

2) Given the authors show they can generate de novo resistance in the gut and identify the resistant clones, this study would be greatly enhanced by testing whether cholestyramine could prevent the emergence of these de novo resistant clones when the mouse is inoculated with only a daptomycin-susceptible strain. This is a more relevant model for this particular bug-drug combination (see comment #4). The authors should briefly comment why they instead chose the mixed R+S inoculation strategy for the experimental design.

[Editors' note: further revisions were suggested prior to acceptance, as described below.]

Thank you for resubmitting your work entitled "An adjunctive therapy prevents enrichment of antibiotic-resistant clones of a colonizing opportunistic pathogen" for further consideration by *eLife*. Your revised article has been evaluated by Dominique Soldati-Favre (Senior Editor) and a Reviewing Editor.

The manuscript has been improved but there are some remaining issues that need to be addressed before acceptance, as outlined below:

As reviewer 2 states clearly below, your handling of samples with low bacterial density needs to be clarified.

Reviewer #2:

The authors establish a mouse model of *E. faecium* gut colonization to test the following hypothesis: Administration of cholestyramine per os, given concurrently with parenteral daptomycin, prevents the enrichment of daptomycin-resistant *E. faecium* in the gut by sequestering daptomycin from the alimentary canal. This is a well-executed study that addresses a major driver of the antibiotic resistance crisis: the off-target effects of antibiotics.

The authors show that parenteral daptomycin selects for daptomycin-resistant *E. faecium* in the mouse gut and, separately, that cholestyramine can sequester daptomycin. Finally, the authors appear to have shown that cholestyramine administration per os can reduce the degree of enrichment of daptomycin-resistant *E. faecium* in the gut. This result provides a powerful proof-of-concept for this strategy with great potential. The authors now also appropriately discuss this general strategy and its major limitations.

The revised manuscript has addressed the major issues raised and I think this paper is much stronger now. However, the revised manuscript now has brand new details that need to be better explained:

1) "For daptomycin-treated mice shedding at least 3 x 10^3^ CFU/10 mg feces at Days 8 and 14…". The statistical analysis included only mice with at least 3x10^3^ CFU per 10mg feces at days 8 and 14. This added detail was not present in the original manuscript and it is not explained why data are being excluded now. There is a mention in the supplemental figure legends of "insufficient bacterial density", but insufficient for what exactly? Given the interpretation of these data relies on a robust statistical analysis and understanding of how the primary data were collected, the authors should more clearly explain the data handling procedure. If proportions were not calculated due to a threshold of <20 CFU/10mg (see #2 below), then what exactly is special about 3 x 10^3^ CFU/10mg?

2) In the test and visualization of the Figure 5A data, the figure legend states data points with <20 CFU per 10mg feces were excluded from the plot, but no explanation is given for why this was done. Presumably this is due to an unacceptable amount of error in calculating proportions? Is <20 CFU also the raw plate count data? If so, this should be explicitly stated in the figure legend so the reader understands why these data were excluded and it can be distinguished from the threshold/rationale for #1 above.

3) Subsection “Adjunctive cholestyramine therapy prevents emergence of daptomycin-resistance”. Why are statistical tests only performed for days 8 and 14 for proportions but for all the days for densities? For example, the plotted proportion data look like an effect may begin to appear by day 4.

4) Subsection “Statistical analysis”; Re: model choice. The authors indicate the proportion values are not bounded by 0 and 1 and so cannot use a binomial model. I defer to the statisticians, but perhaps this can be solved by simply normalizing the data to values between 0 and 1? Basically, just divide all proportion values by the maximum value in the data set, then your values will be bounded by 0 and 1 and all the data will be treated equally in the process.

---

## [Author Response]

Revisions for this paper:1) All reviewers found that the manuscript overstates the generalizability of the findings and the results are more inconsistent than acknowledged. The preventative effect they measured was limited to a narrow range of conditions (Figure 6) and the effect was inconsistent (Supplementary figure 2). First, in the absence of cholestyramine, a relative enrichment of the resistant strain was observed in 4 of the 6 dose-strain combinations (Figure 6A and C), but with an inconsistent dose-response (for Strain PR00708-14, in the absence of cholestyramine, relative enrichment was observed at a daptomycin dose of 100 mg/kg and 200 mg/kg, but not 150 mg/kg). There is significant variability in the pharmacokinetics and pharmacodynamics of antibiotics in humans too, so the narrow dose-effect range observed in this study may be limiting. Also, it may differ substantially for every bug-drug combination. The etiology of this narrow effect window is not known and was not investigated. The authors should discuss this finding as a possible major limitation of the model and potentially to the future application of this overall strategy.

A paragraph addressing these limitations has been added to the Discussion. However, we note that analyzing the data together, as suggested in comment #3 shows that the overall result is robust to many of these details.

2) Additionally, when cholestyramine was administered on the same day as daptomycin, no effect, or perhaps even the opposite effect, was observed (Supplementary figure 2). The data in Supplementary figure 2 is not discussed in the manuscript and is cited in the Materials and methods section with the sentence, "Preliminary experiments were preformed to determine the timing of experimental conditions (Supplementary figure 2)". However, the data presented in Supplementary figure 2 contradicts the main conclusion of the manuscript. The authors should provide further clarity because if one considers all of the data provided, the reproducibility of the major finding described in the manuscript might be significantly diminished.

As suggested by reviewers, the experiment from Supplementary figure 2 has been moved to the main text and included for analysis with the data in Figure 5. Our main findings still hold true in this revised analysis. On the specific issue of timing of cholestyramine administration, we have yet to test that in a single experiment. Formally we have only a difference between Experiments A-C and D, and so cannot attribute that to timing, though that certainly is a factor we need to test going forward.

3) The data in Supplementary figure 2 should be combined with the data in Figure 6, so that the reader can assess all of the data in aggregate. In fact, given bacterial densities were found to not correlate with daptomycin dose (Figure 2—figure supplement 1), it would be reasonable, for each bacterial strain, to combine all doses of daptomycin into a single group. This aggregate data should be presented in the main text and presented with a statistical evaluation of the effect size (comparing “standardl” vs. “chol”) at each time point. This evaluation of the data would be more logical and greatly simplify its presentation. The parsed-out data of individual daptomycin doses could still be presented in the supplemental data section.

This figure (now Figure 5) and the corresponding analysis have been revised as suggested in subsection “Adjunctive cholestyramine therapy prevents emergence of daptomycin-resistance”. The parsed-out data are reported in Figure 5—figure supplements 1-5. This excellent suggestion has really simplified the overall presentation, thank you.

4) The genome sequencing results are unacceptably vague and the requisite data do not appear to be available. You state "We confirmed that the isolated clone (PR00708-14-R) had an increased daptomycin MICc relative to PR00708-14 by broth microdilution (MICc = 12.0). Genome sequencing revealed that PR00708-14 and PR00708-14-R differed by several mutations in hypothetical proteins and noncoding regions, which to our knowledge have not previously been associated with daptomycin-resistance.” Please be specific. Further, this one limited example does not show that daptomycin-resistance evolved in vivo consistently. One could argue that the populations were uncontrolled at some concentrations and grew steadily to higher levels because of some physiological tolerance. Although this is unlikely, it would not be hard to pick a few more clones and confirm in fact that they evolved heritable resistance and identify the causative mutations, which you would report.

Further details have been added to the sequencing results. We have reported MIC data for additional clones from the experiment with PR00708-14, as suggested (subsection “Generation of daptomycin-resistant VR *E. faecium* in the mouse GI tract”).

5) The distinction between preventing de novo emergence of a resistant pathogen and enrichment of a resistant colonizer is important and mechanistically distinct. For mechanisms of antibiotic resistance (such as daptomycin), where point mutations in the genome are causative, resistant clones within susceptible populations are likely to be present at very low frequencies simply due to spontaneous mutation. In this case, this approach does have promise because, in theory, all patients receiving daptomycin would be “at-risk” of developing de novo emergence of daptomycin-resistance. However, for mechanisms of antibiotic resistance that rely on horizontal gene transfer (HGT) or strains with intrinsic resistance, resistant clones do not arise spontaneously from a susceptible population. Instead, a HGT event must take place, or the patient must become newly colonized with that resistant strain. These are relatively rare events. For example, vancomycin resistance in Enterococcus (VRE) is caused by a van gene cassette. This multi-gene cassette does not arise spontaneously. Bacteria get this by HGT and patients get this by being inoculated with VRE in the hospital and, presumably, VRE colonization is sustained in the setting of repeated vancomycin exposures. Thus, a strategy for preventing enrichment of VRE:VSE would only work for patients that are already colonized with VRE. Consider drawing this distinction in the Discussion.

A paragraph addressing these issues has been added to the Discussion.

6) There is insufficient detail of the statistical models used by the authors to assess their validity. The full structure of all reported mixed effects models should be specified. In addition, only p-values for the statistical tests are reported. Effect sizes and their error should also be reported. Full tables of all estimated parameters, including for random effects, from the mixed effects models should be reported.

A new supplementary file (Supplementary file 1) shows the full structure and output for the reported models. In the main text, effect sizes and their error have been added.

7) For analysis of time-series of VRE it is stated that interpolated density values were used. It needs to be clarified why and how this interpolation was carried out, and whether this has inflated the authors' sample size.

For this revision, we have decided not to interpolate the bacterial density data analyzed in this manuscript. The data have been re-analyzed, and the main conclusions have not been affected. The Materials and methods section has been updated accordingly. We believe this simplifies the analysis and avoids inflating the sample size. Thanks for the suggestion.

8) It is simply stated by the authors that model selection was carried out based on AIC. If model selection is to be carried out tables of AIC values should be reported for all fitted models and details of the model selection process need to be given. However, I don't believe model selection is appropriate here. It is well known that model selection based on AIC leads to inflated type I error rates via cryptic multiple hypothesis testing (https://link.springer.com/article/10.1007/s00265-010-1038-5). Statistical results should probably simply be reported from the full model.

As suggested, in this revision we report the full models without model selection based on AIC. The models are reported in Supplementary file 1, and the Materials and methods have been updated.

9) The authors use negative binomial mixed effects models to analyse their data on proportions of resistant bacteria. This appears an odd choice, as a negative binomial distribution is typically used to model abundances. For analysing proportions given from counts a binomial model would be more appropriate. The authors either need to clarify exactly why they have used a negative binomial distribution or re-run these as binomial models.

We have updated the Materials and methods section to better explain our choice of model, and we have added the full model structures to Supplementary file 1. In brief, our proportions were determined by plating samples on agar with and without daptomycin. This gives a count for the total density, and a count for the resistant density. Due to sampling, the resulting proportions were not be bounded by 0 and 1, as required for a binomial distribution. Instead, we used a neg binomial distribution to model the count of resistant bacteria, with total count as an offset.

Revisions expected in follow-up work:1) In these experiments, the resistant clones persisted to day 14, so, further studies are needed to test whether this strategy can interrupt stable colonization, which may be a more relevant end-point than preventing enrichment. However, it should be mentioned in the Discussion that some studies have shown E. faecium outgrowth in the gut precedes clinical infections, as this is pertinent to this strategy and gives this study more merit and relevance.

We have added to the Discussion that colonization often precedes infection. This point is also made in the Introduction.

2) Given the authors show they can generate de novo resistance in the gut and identify the resistant clones, this study would be greatly enhanced by testing whether cholestyramine could prevent the emergence of these de novo resistant clones when the mouse is inoculated with only a daptomycin-susceptible strain. This is a more relevant model for this particular bug-drug combination (see comment #4). The authors should briefly comment why they instead chose the mixed R+S inoculation strategy for the experimental design.

We have added a comment (subsection “Daptomycin treatment enriches for daptomycin-resistant VR *E. faecium* in the GI tract”) explaining our choice of experimental design.

[Editors' note: further revisions were suggested prior to acceptance, as described below.]

Reviewer #2:The authors establish a mouse model of E. faecium gut colonization to test the following hypothesis: Administration of cholestyramine per os, given concurrently with parenteral daptomycin, prevents the enrichment of daptomycin-resistant E. faecium in the gut by sequestering daptomycin from the alimentary canal. This is a well-executed study that addresses a major driver of the antibiotic resistance crisis: the off-target effects of antibiotics.The authors show that parenteral daptomycin selects for daptomycin-resistant E. faecium in the mouse gut and, separately, that cholestyramine can sequester daptomycin. Finally, the authors appear to have shown that cholestyramine administration per os can reduce the degree of enrichment of daptomycin-resistant E. faecium in the gut. This result provides a powerful proof-of-concept for this strategy with great potential. The authors now also appropriately discuss this general strategy and its major limitations.The revised manuscript has addressed the major issues raised and I think this paper is much stronger now. However, the revised manuscript now has brand new details that need to be better explained:1) "For daptomycin-treated mice shedding at least 3 x 10^3^ CFU/10 mg feces at Days 8 and 14…". The statistical analysis included only mice with at least 3x10^3^ CFU per 10mg feces at days 8 and 14. This added detail was not present in the original manuscript and it is not explained why data are being excluded now. There is a mention in the supplemental figure legends of "insufficient bacterial density", but insufficient for what exactly? Given the interpretation of these data relies on a robust statistical analysis and understanding of how the primary data were collected, the authors should more clearly explain the data handling procedure. If proportions were not calculated due to a threshold of <20 CFU/10mg (see #2 below), then what exactly is special about 3 x 10^3^ CFU/10mg?

We have clarified the reasoning for these cutoffs in our figure legends (Figure 5 and Figure 5—figure supplements 1-5), in the Materials and methods, and in the Results. In brief, two different plating methods are presented in this paper, which are now described more clearly in the Materials and methods. 1) Each fecal sample was plated in serial dilution on plates with and without daptomycin to determine densities of total and resistant bacteria, and our detection limit for this plating was 20 CFU/10mg feces. 2) We ran a second assay to more accurately estimate proportions for samples from days 8 and 14. For this assay, 200 CFU from each sample was plated in triplicate on plates with and without daptomycin based on previously determined bacterial densities. This assay required high enough density in the initial sample to prepare six plates at 200 CFU. To satisfy this, we used only samples that had an initial starting density of at least 3 x 10^3^ CFU/10mg feces. Thank you for bringing our attention to this area of confusion, these are important details that are now better communicated.

2) In the test and visualization of the Figure 5A data, the figure legend states data points with <20 CFU per 10mg feces were excluded from the plot, but no explanation is given for why this was done. Presumably this is due to an unacceptable amount of error in calculating proportions? Is <20 CFU also the raw plate count data? If so, this should be explicitly stated in the figure legend so the reader understands why these data were excluded and it can be distinguished from the threshold/rationale for #1 above.

Please see #1. This has been clarified in the revised manuscript.

3) Subsection “Adjunctive cholestyramine therapy prevents emergence of daptomycin-resistance”. Why are statistical tests only performed for days 8 and 14 for proportions but for all the days for densities? For example, the plotted proportion data look like an effect may begin to appear by day 4.

We now include an analysis of proportions across all time points (Model 3, L293-302). An effect does appear by Day 4, and this is now captured in the Results. In addition, we include a separate analysis of Days 8 and 14, for which a different type of plating assay was conducted as described in comment #1 (Model 4, L302-308).

4) Subsection “Statistical analysis”; Re: model choice. The authors indicate the proportion values are not bounded by 0 and 1 and so cannot use a binomial model. I defer to the statisticians, but perhaps this can be solved by simply normalizing the data to values between 0 and 1? Basically, just divide all proportion values by the maximum value in the data set, then your values will be bounded by 0 and 1 and all the data will be treated equally in the process.

Thank you for this suggestion. We have re-run the models as suggested, and we find the results are consistent with our main findings. We have updated the Materials and methods and the models in Supplementary file 1 (Models 3 and 4) accordingly.